# Cell-Penetrating Peptides with Unexpected Anti-Amyloid Properties

**DOI:** 10.3390/pharmaceutics14040823

**Published:** 2022-04-09

**Authors:** Nicklas Österlund, Sebastian K. T. S. Wärmländer, Astrid Gräslund

**Affiliations:** 1Department of Biochemistry and Biophysics, Arrhenius Laboratories, Stockholm University, 10691 Stockholm, Sweden; nicklas.osterlund@dbb.su.se; 2Department of Archaeology and Classical Studies, Stockholm University, 10691 Stockholm, Sweden; seb@student.su.se; 3CellPept Sweden AB, Kvarngatan 10B, 11847 Stockholm, Sweden

**Keywords:** protein aggregation, secretion signal peptide, peptide engineering, drug design

## Abstract

Cell-penetrating peptides (CPPs) with sequences derived originally from a prion protein (PrP) have been shown to exhibit both anti-prion and anti-amyloid properties particularly against prion proteins and the amyloid-β (Aβ) peptide active in Alzheimer’s disease. These disease-modifying properties are so far observed in cell cultures and in vitro. The CPP sequences are composed of a hydrophobic signal sequence followed by a highly positively charged hexapeptide segment. The original signal sequence of the prion protein can be changed to the signal sequence of the NCAM1 protein without losing the anti-prion activity. Although the detailed molecular mechanisms of these CPP peptides are not fully understood, they do form amyloid aggregates by themselves, and molecular interactions between the CPPs and PrP/Aβ can be observed in vitro using various spectroscopic techniques. These initial intermolecular interactions appear to re-direct the aggregation pathways for prion/amyloid formation to less cell-toxic molecular structures (i.e., co-aggregates), which likely is why the disease-inducing PrP/Aβ aggregation is counteracted in vivo.

## 1. Introduction

We here describe designed cell-penetrating peptides (CPPs) which have been shown to display anti-prion and anti-amyloid properties in cell cultures [1,2]. In general, CPPs are relatively short peptides which can transport a molecular cargo across a cellular membrane and into the cell. Such peptides typically display high transport efficiency and low cell toxicity [3]. CPPs can generally be classified by their specific physico-chemical properties such as hydrophobicity and charge. They may also cross the blood–brain barrier, which makes them interesting as possible drug candidates [4].

The CPPs discussed in this review belong to a group of primary amphipathic CPPs, with a hydrophobic N-terminus and a hydrophilic positively charged C-terminus. The peptides were derived starting from the peptide sequences of the N-terminal segments of the mouse prion protein (mPrP_1–28_). This segment consists of a hydrophobic signal peptide (mPrP_1–22_) and a highly cationic hexapeptide sequence (mPrP_23–28_, KKRPKP). The mPrP signal peptide segment was later exchanged into the slightly less hydrophobic signal peptide for a type-I membrane protein found in neuronal cells, i.e., the mouse neural cell adhesion molecule 1 (NCAM1) protein (mNCAM1_1–19_). This yielded the chimeric CPP construct mNCAM1_1–19_-mPrP_23–28_, which we refer to as NCAM-PrP. The peptides’ variants and their properties are summarized in Figure 1A. Both CPP constructs are predicted to be cleaved by signal peptidase I in vivo, resulting in release of the signal peptide and the mPrP_23–28_ peptide.

## 2. Amyloid Processes in Alzheimer’s Disease Involving the Amyloid-β Peptide

Aggregation of the Amyloid-β (Aβ) peptide into amyloid material is associated with neurodegeneration in Alzheimer’s disease (AD). This so-called amyloid cascade hypothesis has been extensively reviewed elsewhere [9]. The Aβ peptide is an amphiphilic peptide with an overall negatively charged N-terminal segment, followed by a hydrophobic segment that originates from a membrane-embedded portion of the amyloid-β precursor protein. Self-interaction between the C-terminal part of Aβ and its central KLVFF motif is believed to be important in the self-assembly process [10]. Figure 1B shows a schematic view of the amyloid aggregation pathway of the Aβ peptide, and the proposed intermediate stages on the way from soluble monomeric peptide into insoluble fibrillar aggregates. Intermediate oligomeric aggregates have been shown to be particularly neurotoxic [11,12]. One suggested mechanism is that such oligomers are capable of disturbing the integrity of cell membranes, for example by formation of transient pore structures [13,14]. Formation of oligomers is greatly enhanced by the presence of fibrillar aggregates, in an autocatalytic cycle [15] (Figure 1B).

Aβ aggregation has been the target for development of therapeutic drugs for AD. Most such efforts have failed, and there are today no generally approved therapies for AD which target Aβ peptides. There has however been some recent success with drug candidates involving the antibodies BAN2401 (Bioarctic AB) and aducanumab (Biogen Inc.), as well as with the small molecule ALZ-801 (Alzheon Inc.). They are all in ongoing or finished clinical trials [16]. All these drug candidates target, at least in part, smaller Aβ aggregates, which further highlights the suspected importance of such intermediate structures in AD.

In vitro studies of Aβ are generally performed on purified recombinant or synthetic Aβ peptides, either the full lengths variants Aβ_1–40_ or Aβ_1–42_ or smaller fragments. The time-dependent aggregation from monomeric peptides into mature amyloid fibrils is often followed using fluorescence spectroscopy with the amyloid-specific dye Thioflavin T (ThT), which significantly increases its intrinsic fluorescence when bound to amyloid fibrils [17]. Other techniques can similarly be used to follow the aggregation process over time, for example by monitoring changes in the Aβ secondary structure with circular dichroism (CD) or infrared (IR) spectroscopy, or by monitoring the size distribution of the Aβ peptide ensemble with dynamic light scattering (DLS) or fluorescence correlation spectroscopy (FCS). Figure 2A,B show examples of kinetic data of Aβ_1–42_, measured using ThT fluorescence and CD spectroscopy in vitro. Such time-dependent studies are commonly used to obtain information about the underlying kinetics of the aggregation processes [18,19]. Kinetic parameters can be supplemented with molecular structures of the peptides and their aggregates. Solution state NMR can reveal the structures of monomeric peptides, while atomic force microscopy (AFM) and electron microscopy (EM) images can reveal the structures of larger aggregates, which might be either fibrillar or non-fibrillar (see examples in Figure 2D).

Detailed studies of the small oligomeric Aβ structures have proven to be more difficult, due to their low abundance and transient nature under normal in vitro conditions. Studies have shown that the maximum amount of Aβ_1–42_ oligomers is reached in the middle of the transition phase during amyloid formation, and that their relative abundance never reaches more than approximately 1.5% of the total peptide molar concentration [15]. Such small populations are usually not distinguishable by methods that detect the average of the entire peptide ensemble. One method that can detect oligomers at low concentrations is mass spectrometry (MS), as in a mass spectrum every molecular species gives rise to a separate signal. The use of soft electrospray ionization and gentle instrument conditions to study protein samples in aqueous solution is usually called native MS, analogous to native gel electrophoresis. Native MS has successfully been used to study Aβ oligomers, their structure, their ligand binding, and their stability upon co-incubation with other molecules [21,22,23].

## 3. Anti-Prion Properties of Signal Peptide Derived CPPs

The discovery of the anti-amyloid CPPs discussed here was partly by serendipity. It had been observed that the mPrP_1–28_ peptide displayed striking sequence similarities with CPPs. In a collaboration with Ülo Langel and his group, we showed that prion peptides with such unprocessed signal sequences are indeed CPPs which can carry large cargoes into live cells [24]. The mPrP_1–28_ peptide could, for example, carry the 67 kDa hydrophilic protein avidin into cells. This was, however, associated with large precipitation as the mPrP_1–28_ peptide was prone to aggregation, especially in the presence of negatively charged lipids. The internalized peptide–complex also seemed to stress the cell, as cell morphology was altered.

It was later observed that the corresponding bovine prion peptide, bPrP_1–30_, was internalized into live mammalian cells by macropinocytosis, initiated by peptide binding to cell surface proteoglycans [25]. The peptide was found to localize to the cytoplasm and not to the nucleus, which could otherwise be suggested by the similarity of the KKRPKP hexapeptide to a nuclear localization signal (NLS). bPrP_1–30_ was found to alter the studied cells just like mPrP_1–28_, indicating some toxicity.

The PrP-derived CPPs were then found to be efficient anti-prion peptides [1,26]. Both mPrP_1–28_ and bPrP_1–30_ could inhibit the conversion between the cellular prion protein PrP^c^ and the misfolded disease (scrapie) form PrP^Sc^, and efficiently counteracted prion propagation in cell cultures. The levels of PrP^c^ in healthy cells were unaffected, indicating a specific effect on the PrP^Sc^ form. The effect was retained, and even enhanced, upon exchanging the PrP signal peptide into the NCAM1 signal peptide [26]. A study based on AFM imaging showed that co-aggregation between our constructs and the prion protein into non-fibrillar/non-amyloid aggregates appears to be an important molecular interaction underlying these results [27]. The NCAM1 signal peptide is shorter and less hydrophobic than its PrP counterpart, which leads to higher solubility and a lower aggregation propensity. This is likely beneficial since aggregation of the PrP-derived signal peptides had been shown to stress the cells.

The anti-prion effect was lost when five or more N-terminal residues were removed from the peptide [26]. There was also no effect for the mPrP_23–28_ segment alone, without any of the signal peptides linked. All this points towards the importance of protein translocation for the CPP constructs to have their anti-prion function. The anti-prion effect was not retained upon exchanging the signal peptide into various other common CPPs, such as the chimeric penetratin-mPrP_23–28_, TAT-mPrP_23–28_, and Transportan-10-mPrP_23–28_ constructs, indicating some specificity for the signal sequence.

The synthetic CPP peptides counteracted prion infection equally well in chiral D-form as in L-form, suggesting that no chiral recognition such as by a receptor is involved in the observed cellular uptake or anti-amyloid function [26]. The D-form of the peptide should promote a longer biological lifetime, which is beneficial for therapeutic purposes. So far, all experiments with our CPP constructs have been carried out in vitro or in cell cultures. Finding out how the constructs perform in vivo, i.e., in animal models or in human patients, is an important task for future research.

No exact molecular mechanism for the anti-prion effect of the CPP constructs has been pinpointed, although co-aggregation between the constructs and the amyloidogenic proteins and peptides appears to be one important aspect [8,20,27]. We hypothesize based on the above-mentioned experimental findings that the signal peptide sequence causes translocation of the peptides into cells where it co-localizes with and binds specifically to PrP^Sc^, through the KKRPKP recognition sequence. The CPP-PrP^Sc^ complex is then unable to bind additional PrP^c^ units, which hinders the conversion from PrP^c^ to PrP^Sc^.

## 4. Anti-Amyloid Properties of Signal Peptide Derived CPPs

Our studies continued from prion infection to more general amyloid effects on living cells. We found that the NCAM-PrP construct could efficiently counteract the cellular toxicity caused by the Aβ peptide, when both peptides were added extracellularly to cell cultures [2]. We also showed that the cationic PrP segment could be exchanged with another cationic hexapeptide (KKLVFF), derived from the Aβ_16–20_ sequence, with similar anti-amyloid effects [2]. The idea behind this new peptide variant, which we labelled NCAM-Aβ, was that the recognition sequence Aβ_16–20_ (KLVFF) is previously known to bind full-length Aβ peptides and affect their amyloid formation in vitro as well as in vivo [28,29]. The KKLVFF sequence also has a strong cationic net charge, similar to the prion-derived KKRPKP sequence previously examined.

The NCAM-PrP and NCAM-Aβ constructs display similar properties in terms of translocation into cells, decreasing Aβ toxicity in cell studies, and decreasing the concentration of Aβ oligomers in vitro as detected by ELISA and dot–blot assays [2]. NCAM-Aβ, however, displayed slightly worse solubility and stability, probably due to the smaller positive net charge of the construct. The NCAM-PrP peptide was also able to suppress Aβ fibrillation at lower CPP/Aβ ratios compared to NCAM-Aβ, further indicating an importance of the positive net charge for the anti-amyloid effect.

In vitro studies of Aβ_1–40_ and Aβ_1–42_ peptide amyloid formation showed that the amyloid process was affected by the presence of the CPPs, as seen by time-dependent ThT fluorescence spectroscopy [8] (Figure 2A). AFM imaging of the aggregation end-point state indicated that the Aβ aggregation process was seemingly redirected to non-fibrillar states by the presence of the CPP (Figure 2D). No clear interactions with the monomeric Aβ peptide were observed in ^15^N HSQC NMR, which instead showed a general drop in signal intensity. A loss in spectroscopic signal upon co-incubation of Aβ and NCAM-PrP was also observed in CD spectroscopy (Figure 2B). No abundant heterooligomer formation was observed using native MS, with only a weak heterodimer being observed (Figure 2C). All this seems to point towards a rapid co-aggregation process where large co-aggregates are formed which precipitate out of solution. Similar non-fibrillar co-aggregates were also observed for the PrP protein together with the NCAM-Aβ construct [27].

A different behavior was observed upon introduction of a membrane-mimicking environment, where loss of NMR, CD, and MS signals was not observed when micelles were included in the samples [8]. Native MS showed that Aβ and NCAM-PrP formed an abundance of heterooligomers in this membrane-mimicking environment, indicating that the two peptides can also interact in a membrane (Figure 2E). Aggregates formed in the membrane are on a size-scale compatible with detection by native MS. This is in contrast to the situation in a pure aqueous solution, where no hetero-oligomerization is observed (Figure 2C). Co-aggregates formed in pure aqueous solution (Figure 2D, right) instead seem to be larger and therefore not detectable by native MS. They also seem to be less soluble, and perhaps also less specific, as seen by the large general losses in signal intensity observed in CD spectroscopy, NMR spectroscopy, and native MS. The co-oligomerization shifted the Aβ oligomer population towards lower oligomerization states (Figure 2F), indicating that the binding of NCAM-PrP interferes with the oligomerization process.

## 5. Discussion

We have here described how CPPs derived from the PrP and NCAM1 signal peptide sequences, together with added cationic hexamer sequences, can decrease the toxicity of PrP and Aβ in cell cultures, as well as modulate their aggregation in vitro. The CPP peptides are made up of a hydrophobic signal peptide part that likely is responsible for translocation, and a cationic short sequence that likely is most responsible for the interaction with the toxic peptides. The exact molecular mechanisms are not known, for example, if the interaction in vivo takes place between Aβ and the intact CPP construct, or if the CPP construct is cleaved upon entry to the cell, thereby releasing the free cationic peptide for further interactions with Aβ and PrP. It is also possible that the interaction could take place in the membrane itself, as both Aβ and the signal peptides are membrane-interacting molecules. Our in vitro results demonstrate direct interactions between small Aβ oligomers and NCAM-PrP in membrane-mimicking micelles. Such interactions could inhibit the formation of e.g., pore-complexes, which have been suggested as particularly toxic states of Aβ aggregation [13,30,31].

The nature of the interaction between Aβ and the CPPs is of interest, as both the PrP and Aβ derived cationic sequences decrease Aβ toxicity. It was even observed that the PrP-sequences were more efficient at counteracting Aβ aggregation compared to the Aβ-sequence, which perhaps indicates that the net positive charge is more important than the specific sequence. One would otherwise expect that the Aβ_16–21_-recognition sequence would interact more specifically with Aβ peptide compared to the mPrP sequence. These speculations on the importance of the positive charge are in agreement with a previous study showing that modifications to the KLVFF segment by adding additional positive K residues increased the segment’s ability to inhibit Aβ aggregation [32]. Another study has similarly shown how increasing the positive charge of hydrophilic proteins lead to enhancement of their ability to inhibit Aβ aggregation [33]. In addition, cationic amphiphiles have been shown to interact strongly with Aβ and inhibit amyloid aggregation [34].

There are recent reports about a relatively short (12 amino acid residues) positively charged peptide, named RD2, which targets Aβ oligomers and shows similar effects as the CPP constructs described here [35,36]. The RD2 peptide has a C-terminal 6xR repeat similar to the cationic hexapeptides presented above, and the D-form of the RD2 peptide showed a beneficial anti-AD effect in animal studies [37]. The suggested model of action is that the RD2 peptide may interact with unstructured Aβ monomers, thereby stabilizing the monomeric state of Aβ [38].

Currently, there is a growing interest in finding sequences of relatively short peptides that may be active against amyloid formation [39,40], where amyloid structures generally are defined as having characteristic cross-β-sheet molecular structures [41]. The Aβ peptide is a typical example of an amyloid-forming pathological peptide. It has a hydrophilic and charged N-terminal segment, followed by a more hydrophobic C-terminal section, which is the basis for formation of the β hairpin structure in the fibrils [10]. The described anti-amyloid peptides have few common features in terms of sequences and potential structures, but certain features seem to be related to the anti-amyloid activity, such as one region having mainly hydrophilic and charged residues and another region mainly having hydrophobic residues [40]. This pattern is in fact similar to generally amyloid-forming sequences. The designed anti-amyloid peptides described here seem to follow this pattern, with the N-terminal segment being mainly hydrophobic and the C-terminal segment being mainly hydrophilic and positively charged. Other short peptides have been found to display anti-cancer activities, with sequences derived from e.g., the azurin protein [42,43]. Here, the ability to enter cancer cells seems to be important for its activity. In general, short peptides, possibly with CPP activities, may be very interesting for covalent or non-covalent addition in the strategic development of therapeutic molecules such as antibodies [44].

## 6. Conclusions

The studies here reviewed, about anti-prion and anti-amyloid effects of certain CPP constructs, suggest that there may be a general principle for how to stop or at least counteract cellular amyloid formation (involving both cell location and cell chemistry). The signal sequence in these constructs may be important for localization of the peptide construct, and the cationic hexapeptide segment may be most important for the anti-amyloid effects on the amyloid and prion aggregation processes.

## Figures and Tables

**Figure 1 pharmaceutics-14-00823-f001:**
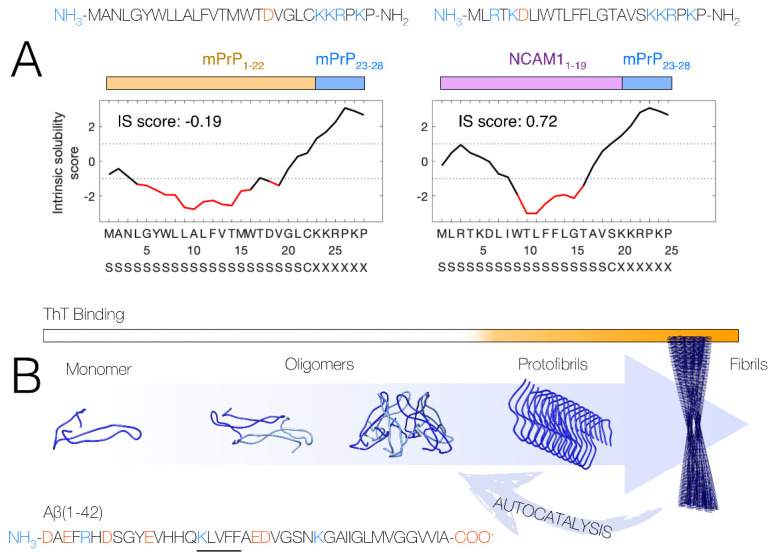
(**A**) Overview of the sequences for mPrP_1–28_ and the chimeric NCAM-PrP construct. Positive charges are colored blue, and negative charges are colored red. The intrinsic solubility score according to the CamSol method [5] over the peptide sequences are shown. Poorly soluble regions are shown in red, indicating the higher solubility of NCAM-PrP compared to the original mPrP_1–28_ peptide. Eukaryotic signal peptide predictions by the SignalP 5.0 method [6,7] are shown at the bottom, with predicted signal peptide segments marked as “S” and predicted cleavage sites by signal peptidase I marked as “C”. The figure is reprinted/adapted from Król, S. et al. The amyloid-inhibiting NCAM-PrP peptide targets Aβ peptide aggregation in membrane-mimetic environments. Science 2021, 24, 102852. https://doi.org/10.1016/j.isci.2021.102852. Ref. [8] under the terms of the Creative Commons CC-BY license. (**B**) Overview of the Aβ aggregation process where soluble monomeric peptides self-assemble into mature amyloid fibrils via intermediate states often termed “oligomers” and “protofibrils”. Oligomers are believed to primarily form in an autocatalytic cycle involving the fibril surface. The fluorescent dye Thioflavin T (ThT) is commonly used to report on the formation of amyloid structures in time-dependent fluorescence spectroscopy experiments.

**Figure 2 pharmaceutics-14-00823-f002:**
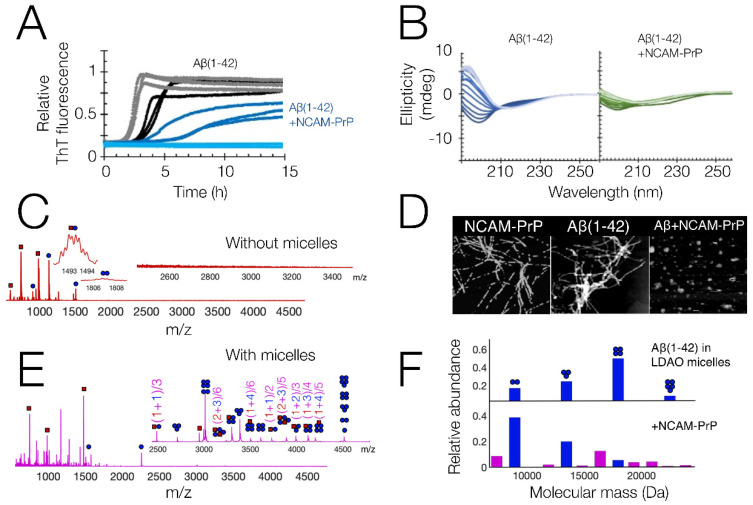
Overview of experimental in vitro results on the NCAM-PrP peptide and its interactions with Aβ_1–42_. (**A**) Time-dependent fluorescence spectroscopy using the amyloid specific ThT dye. Grey/Black: Aβ_1–42_ (with/without 150 mM NaF salt) Light blue/Dark blue: Aβ_1–42_ + NCAM-PrP at 1:1 molar ratio (with/without 150 mM NaF salt). 5 μM peptides in 20 mM NaP buffer pH 8. (**B**) CD spectra recorded at multiple time-points (lightest trace = 0 h, darkest trace = 4 h), of Aβ_1–42_ alone (blue) and Aβ_1–42_ + NCAM-PrP at 1:1 molar ratio (green); (**C**) electrospray ionization mass spectrum of Aβ_1–42_ + NCAM-PrP at 1:1 molar ratio, at native conditions. Aβ_1–42_ species are shown as blue circles, NCAM-PrP species are shown as red squares; (**D**) Left: 1 × 1 µm AFM image of 10 µM NCAM-PrP peptide incubated for 26 h in PBS buffer, pH 7.4 at 42 °C [20]. Reprinted (adapted) with permission from Pansieri, J. et al. Pro-Inflammatory S100A9 Protein Aggregation Promoted by NCAM1 Peptide Constructs. ACS Chem. Biol. 2019, 14, 1410–1417; Copyright 2019 American Chemical Society. Middle, and Right: 2 × 2 µm AFM images of 5 µM Aβ_42_ peptide, incubated without (middle) or together with 5 µM NCAM-PrP peptide (right), for 15 hrs in 20 mM NaP buffer, pH 8 at 37 °C [8]. (**E**) electrospray mass ionization mass spectrum of: Aβ_1–42_ + NCAM-PrP at 1:1 molar ratio, at native conditions, similar conditions as in C, but with 4 mM LDAO micelles to mimic a membrane environment. The peptide species were removed from the micelles using collision induced dissociation; (**F**) overview of how the distribution of Aβ_1–42_ oligomers in the LDAO micelles shifts upon co-incubation with NCAM-PrP. All results are adapted and reprinted from reference [8], under the terms of the Creative Commons CC-BY license, unless otherwise is stated.

## Data Availability

Not applicable.

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
