# Peer review of "Cell-Penetrating Peptides with Unexpected Anti-Amyloid Properties"

_pharmaceutics, 2022, doi:10.3390/pharmaceutics14040823_

Round 1
Reviewer 1 Report
In this review the authors are describing Cell-Penetrating Peptides (CPPs) with unexpected properties; concentrating CPPs with sequences derived originally from a prion protein (PrP) that have been shown to exhibit both anti-prion and anti-amyloid properties particularly against the amyloid-β (Aβ) peptide active in Alzheimer´s disease.
This review is interesting; however, it is very focused in its view, and as such is for a very limited professional audient.
The manuscript is actually a summary of authors’ research during the years and almost completely self-referred. It is highly recommended to open up the scoop of the review for additional CPPs with special properties; for example describing and referring to CPPs that have anti-cancer properties, such as p18 a domain derived from the protein azurin. It will be interesting to compare the sequence of such CPPs. Any other examples for CPPs with special properties are as also welcomed. This broader view will improve the review for many more scientists and physicians.
Also, the authors should expand their discussion regarding the therapeutic possible applications of these peptides behind AD and prion diseases, i.e. other aggregation-diseases.
In this review the authors are describing Cell-Penetrating Peptides (CPPs) with unexpected properties; concentrating CPPs with sequences derived originally from a prion protein (PrP) that have been shown to exhibit both anti-prion and anti-amyloid properties particularly against the amyloid-β (Aβ) peptide active in Alzheimer´s disease.
Author Response
Please see response to editor:
We hereby submit an updated version of our manuscript “Cell-Penetrating Peptides with unexpected anti-amyloid properties”.
The opinions by all reviewers were similar: they considered the original manuscript to be too narrow and the scope should be widened. We have now increased the discussion section considerably, and have added more general information about the use of small peptides as anti-amyloid therapeutics. We have increased the number of references from 37 to 44, and all new references are from outside and not from our own work. We have also included a couple of references about anti-cancer activities of short peptides. However, we still conclude that our small anti-amyloid peptides are developed from Cell Penetrating Peptides, and consider this to be an important aspect of the review. We would like to point out that our figures are modified versions of our previously published figures, which we have published open access under copyright licences that allow the figures to be re-published. Thus, there are no copyright issues with our figures. We understand that our publishing fee will be covered by Stockholm University.
Yours sincerely,
Prof. Astrid Gräslund
Reviewer 2 Report
In this work the authors summarize and discuss their recent findings about CPPs derived from prion signal peptides having anti-prion and anti-amyloid properties. The paper is clearly structured and well-written. Details behind the activity mechanisms of those peptides are discussed.
This is an interesting piece of work that can be published as is. As minor point I only ask about the future application in vivo of those peptides. The studies presented are done in vitro, but eventually, the authors can give some outlook in this direction.
Author Response

(The authors gave the same response as above.)

Reviewer 3 Report
The manuscript is well written and the topic is interesting. However, this is not a review but a brief account on the authors own work. No less than 14 citations out of 37 are from the author's own group. I think to consider this manuscript as a review in Pharmaceutics, the scope should be substantially widened, for instance including a perspective on anti-amyloid peptides.
Author Response

(The authors gave the same response as above.)

Round 2
Reviewer 1 Report
The corrections incorporated are appropriate and appreciated.